# Carotenoid Yeasts and Their Application Potential

**DOI:** 10.3390/foods14111866

**Published:** 2025-05-24

**Authors:** Ewa Kulczyk-Małysa, Elżbieta Bogusławska-Wąs

**Affiliations:** Department of Applied Microbiology and Human Nutrition Physiology, Faculty of Food Sciences and Fisheries, West Pomeranian University of Technology in Szczecin, Papieża Pawła VI 3, 71-459 Szczecin, Poland; elzbieta.boguslawska-was@zut.edu.pl

**Keywords:** carotenoid microorganisms, metabolism pathways, biological activity, bioactive additive

## Abstract

Carotenoids are part of a diverse group of isoprenoid compounds. Due to the many properties they possess, they may become an alternative to synthetic additives in various industrial sectors. The increase in consumer demand and awareness determines research into extracting them from plants, algae and microorganisms. The extraction of carotenoids from plants is an inefficient method and generates additional production costs. On the other hand, the carotenoid potential of microorganisms, especially among yeasts, has not been fully exploited. The diversity of yeast species and strains influences the extraction of many fractions of carotenoids, including the less known ones such as thorulene and tholuradine. The developed adaptability of yeast enables the optimisation of their culture, which facilitates the understanding of their metabolic pathways. At the same time, the coordination of carotenoid and lipid synthesis may prevent their degradation or the loss of their bioactive properties. Application research has been conducted mainly in the feed industry, where their colouring and antimicrobial or immunomodulating properties are used. In the medical and pharmaceutical fields, there is not much research due to safety restrictions and the necessity of the high purity of the fractions. This review also highlights the overlooked aspect of carotenoids’ biodegradability, which is required to exploit the bioactive properties of microbial carotenoids.

## 1. Introduction

Food additives are currently an integral part of the processing steps leading to the final product. They have been defined as substances that are not consumed as food and are usually not used as a characteristic ingredient of food, regardless of their nutritional value. Among the approved compounds, synthetic and natural food additives are distinguished. The increasing expectations of consumers have led to a conscious choice of buying products with a composition based on natural ingredients, including containing functional substances [1,2]. Carotenoid compounds represent one of these. There is a growing trend in the global market for their use in various food industries. According to The Global Market for Carotenoids report, the global carotenoids market should reach USD 2.7 bn by 2027, with a compound annual growth rate (CAGR) of 5.7% during the forecast period from 2022 to 2027. The data reported clearly show an increasing trend in the demand for carotenoids, including the carotenoid fractions of microbial origin [3].

In accordance with Regulation (EC) No 1333/2008 of the European Parliament and of the Council from 16 December 2008, a list of four carotenoid additives (E 160a) was published. These compounds have been classified into four main groups, depending on their source: E160a (i)—chemical synthesis, E160a (ii)—plant carotenes, E160a (iii)—biosynthesis by *Blakeslea trispora* and E160a (iv)—carotenes from algae, extracted from *Dunaliella salina* [4]. However, Commission Regulation (EU) No 231/2012 from 9 March 2012 contains specific quality criteria for food additives, including natural carotenoids. Among the natural colours, in addition to carotenoids, compounds derived from the female insect *Dactylopius coccus Costa* are used (E 120). The use of insect-derived cochineal is unacceptable to many consumers, and products containing this compound are avoided. Consequently, *Blakeslea trispora* pigments are more commonly used, with guidelines on the degree of purity and microbiological quality of the natural carotenoids. These restrictive criteria are intended to ensure their safety when applied in food products [5]. Among the authorised additives, as can be observed, there are only two of microbiological origin. However, the potential of carotenoid microorganisms is significantly greater. Consequently, there has been increasing scientific interest in their possible applications [3]. Yeasts are a particularly interesting area of research. As euryecological forms, they are characterised by a natural ability to adapt to changes in ecosystems. In many cases, this promotes the expression of genes responsible for generating the synthesis of many bioactive compounds, including carotenoid compounds. Among yeasts, only two species have GRAS status: *Yarrownia lipolitica* and *Saccharomyces cerevisiae*, which are not carotenoid yeasts. Moreover, some representatives of carotenoid yeasts of the genus *Rhodotorula* are opportunistic pathogens. However, when applying carotenoid fractions extracted from carotenoid yeasts themselves, the focus should be on obtaining fractions of adequate purity and determining their cytotoxicity [3].

Therefore, the aim of this paper is to review the current knowledge on a group of bioactive carotenoid compounds, with a particular focus on yeasts as their producers. Moreover, the potential use of carotenoids in the food and medical industries as valuable compounds with health-promoting properties will be presented. The bioavailability of microbial carotenoids and their effect on the host gut microbiota will be discussed for the first time.

## 2. Characteristics of Carotenoids

Carotenoids are 40-carbon terpenoids, included in the isoprenoid group of compounds [6,7]. They are characterised by the presence of a conjugated system of double bonds, the polyene system of which typically comprises up to 15 of these bonds. This structure is determined by the presence of absorption spectra in the visible light range from 400 to 550 nm and their photochemical properties [8]. The double bond region represents a critical point in the carotenoid molecule on which their degradation depends [9]. Considering their properties, carotenoids are highly hydrophobic molecules. As a result, they dissolve in oil phases and organic solvents and do not dissolve in water [7]. Due to their ability to bind oxygen radicals, these compounds exhibit antioxidant properties. A special attribute of these compounds is their orange-red colour, which imparts a specific colour to the colony [10]. Different classification criteria for carotenoid compounds are adopted. Regarding their chemical properties, carotenoids are generally divided into xanthophylls and carotenes, which differ by the presence of oxygen in the molecule. Xanthophylls are made up of hydrogen, carbon and oxygen, which can be found in the form of hydroxyl, epoxide or oxide groups. Xanthophylls include lutein, astaxanthin, canthaxanthin and zeaxanthin, among others. In contrast, the structure of carotenes involves carbon and hydrogen forming a polyunsaturated cyclic (α-carotene, β-carotene, γ-carotene) or acyclic (lycopene) chain [11,12]. According to their ability to produce the vitamin, a distinction is made between carotenoids that are provitamin A such as α- and β-carotene and those that are not provitamin A, including lycopene and zeaxanthin [7,13,14].

## 3. Carotenogenic Microorganisms

Carotenoids may be synthesised by chemical or natural means. The commercial production of carotenoids is mainly based on chemical synthesis, which accounts for more than 73% of the market demand. However, synthetic carotenoids, especially in high doses, could be harmful to human health. Furthermore, the waste generated in their production process poses a threat to the environment. This has become the reason for the development of new technologies for extracting carotenoids from natural sources—plants, animals and microorganisms, whose widespread use is still limited. Carotenoids of plant origin, known as phytonutrients, are biologically active compounds that are appreciated in the food industry primarily for their application potential. Their increasing use in the cosmetic and pharmaceutical sectors is also indicated [9]. The extraction of carotenoids from vegetables is related to the use of expensive extractants and their low stability. Seasonality of crops and geographical variability are also a real problem, which is controlled by maintaining adequate light or temperature in greenhouses. However, such solutions generate additional costs, reducing the profitability of the process [15]. In contrast to plant carotenoids, the use of microbial synthesis allows natural compounds to be obtained while maintaining cost-effective production. In addition, the potential to increase the yield of microbial metabolite synthesis using fermentation processes in bioreactors with optimised control of the cultivation processes is indicated [16]. The biotechnological production of carotenoids can be achieved using algae, bacteria and fungi (Table 1). Currently, the algae *Hametococcus pluvidis* and the algae of the genus *Dunaliella* spp. are used on an industrial scale, and are primarily producers of astaxanthin and β-carotene. Regarding moulds, carotenoids are obtained from the culture of *Blaklesia trispora* (β-carotene). On the other hand, *Monascus* sp., *Rhodotorula tainanensis* and *Phaffia rhozdyma* are representatives of yeasts whose production is also being commercialised. However, the carotenoid potential of microorganisms has not been fully exploited. This is particularly the case with yeasts, which show the ability to synthesise compounds that are specific and rarely found in the environment [17,18].

In the microbial world, the ability to synthesise carotenoid compounds is common. However, it is most frequently considered because of evolutionary adaptation in response to environmental phototoxicity. In the case of yeasts, the genera *Sporobolomyces*, *Sporidiobolus, Phaffia, Rhodosporidium* and *Rhodotorula* are the most identified representatives of carotenoid fungi [1,30].

Representatives of the genus *Rhodotorula* are included in the so-called ‘oleaginous yeasts’, which show the ability to produce lipids, exopolysaccharides and carotenoids. They represent the largest group of yeasts with a naturally genetically determined potential to synthesise specific carotenoid fractions [1]. In optimised conditions, depending on the strain, compound accumulation efficiencies range from 93.9 µg/g dry weight to 16.9 g/L. *Rhodotorula* sp. are common environmental yeasts whose phenotypic characteristics differ significantly. One of the factors for optimising the production process is the C/N ratio. An increase in the ratio generally leads to an increase in the efficiency of carotenoid synthesis, which may not be a constant feature. This is demonstrated by studies carried out by Kulczyk-Małysa and Bogusławska-Wąs [36], which isolated *Rhodotorula* strains that did not confirm this relationship. In controlling the synthesis process, the percentage of sodium chloride is significant. In a study by Kanzy et al. [43], it was confirmed that the highest concentration of carotenoids was obtained in the medium with the addition of whey containing 3% NaCl. In the case of the yeasts *Sporobolomyces* sp. and *Sporidiobolus* sp., a higher efficiency of β-carotene production was determined in whey containing 5% NaCl [44]. The indicated yeast types would also be efficient producers of EPSs (exopolysaccharides). On an industrial scale, exopolysaccharides are used in the food industry to concentrate food products. By regulating the glucose content of the medium to 120 g/L, their yield could be increased [45]. The same relationship was shown by Cabral et al. [46] for the production of carotenoids by the species *Sporidiobolus pararoseus*. The biotechnological use of *Sporobolomyces* sp. and *Sporidiobolus* sp. is associated with the preceding mutagenesis processes. The introduction of hydrogen peroxide into the culture increases the concentration of ROS (reactive oxygen species). The resulting stressful conditions might stimulate β-carotene synthesis [47]. A further representative of carotenogenic yeast is *Phaffia* sp. A characteristic feature of the representatives of this genus is the existence of two amorphous forms: the asexual *P. rhodozyma*, and the sexual form *Xanthophylomyces dendrorhous*. Both forms are used to produce carotenoids. Astaxanthin is the dominant carotenoid fraction that is used in the food and feed industry. Research is mainly focused on optimising its production [48]. In the case of *Phaffia* sp., a yeast fermenting glucose, maltose, sucrose or raffinose, the most efficient synthesis is achieved in a culture containing up to 50 g/L glucose. The use of higher concentrations results in a decrease in its synthesis, as shown by Liu et al. [49] in the optimisation of *X. dendrorhous* cultures. A similar relationship is observed for lipid production by yeast strains belonging to *Rhodosporidium*. In this instance, an increase in glucose concentration determines an increase in yeast dry weight, while the efficiency of lipid synthesis decreases. *Rhodosporidium* sp. demonstrates the ability to accumulate up to 50–70% lipids (*w*/*w*) of cell dry weight. The composition of the lipids produced by yeasts of the genus *Rhodosporidium* determines the results of many studies enabling the introduction of a vegetable oil substitute into the consumer market [50].

### Synthesis of Carotenoid Yeast Metabolites

In consideration of the properties of the compounds synthesised by yeast and the potential for their application in various sectors, terpenoid biosynthetic pathways have been at the centre of scientific interest. Terpenoids represent the largest group of natural compounds with bioactive properties [51]. As the result of reactions, i.e., where carotenoids (tetraterpenoids) and lipids are produced, the optimisation of their production process on an industrial scale is still being improved. Control of the pathways for the synthesis of these compounds occurs at the level of the mevalonate (MVA) and/or 2-C-methyl-d-erythritol 4-phosphate (MEP) pathways. The MEP pathway involves subsequent reactions occurring in Gram-negative bacteria, microalgae and plants. In eukaryotes, archaea and Gram-positive cocci, the biosynthesis of terpenoid compounds occurs through the MVA pathway, which begins with acetyl-CoA. In yeasts, carotenoids are synthesised via the mevalonate pathway (Figure 1) [52,53].

Carotenoid production begins with the assimilation of substrate compounds. The primary simple carbohydrate is glucose, which is converted in the EMP (Embden–Meyerhof–Parnas) pathway occurring in the cytoplasm. The reaction products include pyruvate molecules, which are transported into the mitochondrion and incorporated into the tricarboxylic acid (TCA) cycle. The non-ionised form of citrate, i.e., citric acid, is transported from the mitochondrion to the cytoplasm and then contributes to the formation of acetyl-CoA through ATP citrate lyase [59]. In yeasts, acetyl-CoA is incorporated into the MVA pathway and converted to a key precursor molecule, i.e., isopentenyl pyrophosphate (IPP) [60]. The three IPP molecules undergo condensation to geranyl pyrophosphate (GPP), leading to a direct precursor in carotenoid biosynthesis, the geranylgeranyl diphosphate C20 (GGPP) molecule. Phytoene as the first carotene in the described pathway is produced by phytoene synthase (PSY). Further conversion in the next step leads to the formation of lycopene [57]. The cyclization of lycopene at the ends of the molecule chain results in the synthesis of γ-carotene (one β-ionone ring) and β-carotene (two β-ionone rings). In the subsequent process of dehydration, thorulene is being formed. On the other hand, thorularodine is synthesised after the hydroxylation and oxidation of thorulene, which allows for the incorporation of a carboxyl group. The yellow pigment, astaxanthin, is formed by the hydroxylation and ketolation of β-carotene (Figure 2). The diversity of the conjugated bonds and functional groups in these compounds determines their antioxidant activity and the occurrence of a variety of colours, from red to yellow [61].

Lipids, like carotenoids, are terpenoid compounds whose synthesis begins with IPP molecules [51]. The process of lipid synthesis could occur de novo or ex novo (from hydrophobic substrates). De novo lipid synthesis is an anabolic biochemical process, where the fundamental units of biosynthesis are acetyl- and malonyl-CoA. On the other hand, in the ex novo process, external substrates such as alkanes, fatty acids and oils are first degraded and hydrolysed. After these processes, the lipids are then transported into the cell. The overproduction of lipids in the de novo pathway occurs when nitrogenous compounds are limited in the culture. This is associated with the activation of AMP deaminase, which degrades AMP (adenosine monophosphate) with the release of ammonium ions [65]. Disruption of the TCA cycle prohibits the balance of the citrate concentration in the yeast mitochondrion, which is transported into the cytosol and degraded to acetyl-CoA [61]. Acetyl-CoA is carbonylated to malonyl-CoA and the fatty acid enzyme complex (FAS) enables the production of triacylglycerols. The analysis and optimisation of ex novo pathways currently results from the possibility of using waste substances to synthesise microbial lipids, so-called SCOs (single-cell oils). Then, substrates such as glycerol extracted from biodiesel production are emulsified by liposan molecules and released by oleogenic yeast [62]. After a phase of external hydrolysis conditioned by extracellular lipases, fat molecules are transported into the cell and integrated into lipid metabolism pathways. Accumulated fatty acids in lipid bodies under stressful conditions can be incorporated into the β-oxidation pathway, which leads to the release of acetyl-CoA, a substrate for lipid and carotenoid synthesis [65] (Figure 3).

The control of lipid synthesis pathways has a close relationship with carotenoid biosynthesis. In addition to regulating basic factors such as temperature, pH, C and N sources, C/N or the aeration rate, the optimisation of production can also occur at the level of altering metabolic pathways. The main precursor in the lipid and carotenoid synthesis pathways is acetyl-CoA. Due to the two lipid synthesis pathways, two strategies to increase the synthesis efficiency of the mentioned metabolites are considered. Increasing carotenoid synthesis by increasing lipid synthesis de novo has not been successful. The reason for this is that there is competition between the lipid synthesis pathway and the MVA pathway. This results in the observed decrease in product yields from both pathways. Considering the properties of carotenoids and their ability to dissolve in oil phases, running parallel synthesis processes may affect the stability of carotenoid fractions [68]. The approach of activating ex novo lipid synthesis is determined by supplying fatty acids such as palmitic acid to the culture medium. Excess fatty acids are stored in fat bodies. Their release or the direct incorporation of FFAs into peroxisome-mediated metabolism—the β-oxidation pathway—increases acetyl-CoA production allowing an increase in isoprenoid synthesis (Figure 3) [69,70]. The metabolic pathways of carotenoid yeast can also be controlled by co-culturing with symbiotic microorganisms—lactic acid bacteria. The growing interest in the use of industrial wastes determines processes that increase the assimilation of the components contained in these wastes. Conducting co-culturing is mainly related to the desire to use whey. The main carbon source in whey is lactose [71]. A lot of yeasts have no ability to assimilate this disaccharide, which prevents them from growing. The introduction of lactic acid bacteria allows lactose to be fermented and degraded into yeast-assimilable monosaccharides. This approach may allow the increased productivity of the culture through the availability of glucose to mediate the formation of acetyl-CoA [72].

## 4. The Application of Carotenoid Yeast

### 4.1. Colouring Properties

The directions of the food industry’s development are determined by the specificity of the industry, which is significantly influenced by the consumer. In addition to aspects related to taste, appearance or smell, customer satisfaction is also determined by the product’s impact on health. Many synthetic compounds with a colouring function have been introduced to the food additives market. However, the increase in observed negative effects has led to a search for new sources of pigments suitable for use in the food industry. Biological pigments, including microbial pigments, are the solution to the new market needs [73]. The use of microbiologically sourced carotenoids requires attention to be paid to their stability in the food product. Tetraterpenoids are compounds sensitive to technological processes that may result in degradation. Understanding the mechanism of carotenoid oxidation would therefore be crucial in optimising the stability of bioactive components. Factors that degrade the polyene chain include high temperature, singlet oxygen interaction, photodegradation and isomerisation [7]. Maintaining the stability of these compounds is therefore a challenge for which food producers are looking for new solutions. The use of carotenoid yeast is mainly related to its colouring properties. Colour has been identified as one of the most important attributes of food products determining consumer choice. Therefore, Sharma and Ghoshal [74] have conducted research to determine the potential of the extracted pigment from *R. mucilaginosa*. The researchers added it to two different confectionery products (hard candy and jelly) at concentrations of 0.1, 0.15 and 0.2% (*w*/*w*). As the colour intensity increased, there was an increase in the carotenoid content and antioxidant activity of the products obtained. The results showed that there was a linear relationship between carotenoids and antioxidant activity. Furthermore, the antioxidant activity of the jelly increased from 42.80 to 86.19% at levels of pigment addition from 0.0 to 0.20%. The bio-dye produced from *R. mucilaginosa* is a heat-stable mixture of several carotenoids, mainly torularodine, β-carotene and thorulene, enabling it to combine the functions of natural colour and antioxidants. One of the most important characteristics of poultry quality, indicated regardless of the class, is the colour of the meat, carcass and egg yolk. To obtain these quality characteristics, pigments, including carotenoids, are added to the feed. Currently, it is mainly synthetic carotenes that are used. However, those of microbial origin have additional bioactive properties that may have a beneficial effect on the animals. In broiler feed, up to 80 mg/kg of pigment additives are allowed. Restrictive standards are associated with the use of canthaxanthin, where the addition should not exceed 8 mg/kg for laying hens and 25 mg/kg feed for broilers [75]. The addition of red yeast, which produces carotenoid fractions such as β-carotene, makes it possible to increase the amount of retinol provided in the poultry diet (0.66 μg of β-carotene represents 1 U of retinoic acid). Astaxanthin supplementation has also been used for obtaining an appropriate colour, not only for the egg yolk, but also for the eggshell. Therefore, Zhu et al. [76] conducted an experiment for 6 weeks by feeding suitably formulated feeds together with the addition of red yeast in the range of 0–1600 mg/kg. The astaxanthin content was at a level of 0.96–1.92 mg/kg. According to the hypothesis, as the addition of yeast increased, the level of carotenoids and thus the colour intensity of the egg yolk increased.

In aquaculture, research is being conducted into the use of carotenoids contained in yeast as compounds to promote favourable pigmentation in fish and shrimp. Due to the desire to improve the colour of salmon, among others, the use of natural carotenoids, mainly astaxanthin, is increasingly favoured. Salmonid fish and most animals have no ability to synthesise this compound. The preferred pigmentation of salmonids is therefore mainly dependent on exogenous sources supplied with the feed. Johnson et al. [77] used *P. rhodozyma* to improve the colour of rainbow trout (*Salmo guirdneri*). After 48 days of feeding the fish, astaxanthin was determined to be at a level of 10.6 pg/g of flesh, and also acquired a favourable pink colour. Ueno et al. [78] isolated the carotenoid strain *Rhodotorula dairenensis* Sag 17 from an aquatic environment. After obtaining a coloured preparation, they applied it for 3 months in fish cultures of tilapia (*Oreochromis niloticus*) and black carp (*Cyprinus carpio*). The authors showed an increase in carotenoid content in fish tissues with increasing culture time. An improvement in fish pigmentation was observed in tilapia, while it was not as apparent in black carp due to their dark skin colour.

### 4.2. Antioxidant Properties

The antioxidant properties of carotenoids have been exploited in the feed sector by An et al. [79]. The addition of *Xanthophyllomyces dendrorhous* reduced lipid peroxidation in broiler meat stored for 4 weeks. The inhibition of lipid peroxidation also reduced aldehyde production, preventing the unpleasant taste of meat. In addition to the mentioned yeast, Sun et al. [80] carried out an enrichment of laying hen feed with *Rhodotorula mucilaginosa* and its fermentation products. Feed supplementation of 0.5, 2.5 and 12.5% (*w*/*w*) had no effect on egg shape, strength or shell thickness. In contrast, an increase in egg yolk colour intensity was observed, which correlated with higher carotenoid concentrations.

Most of the research conducted in the food industry is concerned with obtaining a formulation that exhibits antioxidant and antimicrobial properties. The use of carotenoids to quench singlet oxygen makes it possible to prevent lipid peroxidation in food products and maintain the high quality of stored food products [81]. Manimala and Murugesan [82] determined the antioxidant and antimicrobial activity of the pigment extracted from *Sporobolomyces* sp. The experiments confirmed a DPPH radical scavenging level of 75.04%, with a formulation concentration of 100 μg/mL. This suggests that the compounds can be used not only as a natural dye, but also as an antioxidant compound. Similar research was carried out by Gramza-Michałowska and Stachowiak [83] using carotenoids produced by *P. rhodozyma* CBS 5626. The linoleic acid emulsion system was used to evaluate the antioxidant properties of the extracts. The results of the microbial comparison of the formulation with synthetic astaxanthin indicated a formulation with *P. rhodozyma* with three times higher antioxidant efficiency. In contrast, García-Béjar et al. [84] isolated indigenous yeasts from fermented sausages and demonstrated the presence of *R. mucialginosa* in game meat products. They then examined the potential of the yeast to inhibit lipid peroxidation, which was approximately 50%.

Antioxidant compounds have an important role in regulating cell apoptosis processes and preventing the photo-ageing of cells which are considered to be the cause of skin firmness loss [85]. The natural carotenoid fractions β-carotene, lycopene and astaxanthin are known as photoprotectors and their addition to UVA protection creams can effectively replace their synthetic counterparts. However, the use of microbially derived carotenoids in dermocosmetics is still under-researched and requires the development of a safe method for their extraction and application. Furthermore, scientific evidence suggests that antioxidants reduce the risk of chronic diseases, including cancer and heart disease. According to the scientific evidence, their main role is to scavenge free radicals, which are commonly present in biological systems. The consequence of their presence is the oxidation of cellular components, i.e., nucleic acids, proteins, lipids or DNA, which can lead to the development of diseases [86].

### 4.3. Antimicrobial Properties

Preventive interventions against microbial infections are increasingly being introduced in animal breeding. One approach involves promoting the development of a favourable gut microbiome. An example of a sector where infections cause economic losses is white shrimp (*Litopenaeus vannamei*) farming. The common practice of antibiotic therapy is associated with the detection of residues of these compounds in the environment. The use of probiotics, including *Rhodotorula paludigena* CM33 by Sriphuttha et al. [87] confirmed their antioxidant properties. Moreover, the addition of yeast at a level of 5% influenced the biodiversity of the shrimps’ gut microbiome. Analysis of the effect of yeast on Vibrio parahaemolyticus cells, the most common cause of infection in culture, confirmed the inhibition of the growth of this pathogen. The same effect was obtained by Wang et al. [88] using *Rhodotorula benthica* strain D30 isolated from *Apostichopus japonicus*. Commonly known as sea cucumber, it is cultured for its high protein content and suitability for the production of protein powder preparations. A problem for this species includes the occurrence of skin ulcers causing losses in breeding and chronic skin diseases in individuals. The use of this yeast strain has reduced the occurrence of diseases in *Apostichopus japonicus*. According to the authors, this is becoming a good alternative to antibiotic therapy. An improvement in growth rate and an increase in digestive enzymes in A. japonicus is also indicated. On the other hand, in poultry farming, feed supplementation through the addition of *Phaffia rhodozyma* has enabled an increase in the diversity indices of the caecal microbiome of laying hens. The presence of multiple bacterial species has contributed to a reduction in the proliferation of pathogens, enabling the prevention of the occurrence of diseases in birds [89].

Foodborne diseases (FBDs) caused by foodborne pathogens represent a significant and increasing public health problem. *Staphylococcus aureus*, *Listeria monocytogenes*, *Salmonella typhimurium* and *Escherichia coli* are among the important microorganisms causing FBDs. A study by Manimala and Murugesan [82] confirmed the antimicrobial properties of *Sporobolomyces* sp. pigment against selected Gram-positive and Gram-negative bacterial species, including, among others, *E. coli*, *S. aureus* and *Bacillus subtilis*. Similar effects were obtained by Naisi et al. [90] using extracted carotenoids from *R. glutinis*. The preparation was effective in inhibiting the growth of pathogenic strains of Staphylococcus isolated from the milk of cows suffering from mastitis and *S. typhimurium* isolated from poultry. Carotenoids from *R. glutinis* were also shown to effectively inhibit biofilm formation. The mechanism of action of the extracted compounds was based on the inhibition of Quorum Sensing (QS) by reducing the expression of genes controlling the QS system and preventing cell aggregation. The carotenoids exhibited antimicrobial activity with a low level of toxicity to eukaryotic cells. This indicates the potential of the formulation as a natural preservative in various food materials.

In medicine, an important application of appropriately purified carotenoid fractions is their use as antimicrobial products. This applies mainly to titanium medical products relevant to transplants and implants for preventing bacterial infection. The major problem during implant application is the adhesion of microorganisms on the surface of medical plastics and their proliferation. Thus, scientists have been undertaking research into the application of, among other things, thorularodine produced by yeast in films covering implants [91]. The possibility of using carotenoids as protective substances for surgical instruments against the colonisation of antibiotic-resistant bacteria is also indicated, which creates another area of use. Another aspect is cell mutagenesis, which is significantly influenced by ROS. Sinha [92] demonstrated the selective cytotoxicity against breast cancer cells of extracted carotenoids from the *R. toruloides* strain ATCC 204091. In addition, the authors showed selective antiproliferative activity against molecular and cell signalling pathway proteins produced by cancer cells. This indicates a potential higher efficacy in anti-cancer therapy. Another interesting experiment carried out by Sinha et al. [93] was the evaluation of the antimalarial activity of extracted carotenoids produced by strains of *Rhodotorula* sp. *Plasmodium falciparum*-causing malaria was effectively inhibited at stages prior to their invasive form. The authors demonstrated a 96.9% inhibition of the parasite at a concentration of 10^−4^ μg/μL of carotenoid extract. Furthermore, the extracted carotenoid was non-toxic to erythrocytes and HepG2 cells, but active against the 3D7 malaria parasite strain *P. falciparum*.

A popular use of carotenoids is also oral supplementation. When taken in supplement form, β-carotene supports sun protection by preventing sunburn and photo-ageing. Production of this fraction occurs mainly through chemical synthesis. There are also supplements based on natural sources obtained mainly by extraction from plants or algae—*Dunaliella salina*. Preparations obtained by fermentation of *Blakeslea trispora* may also be used by humans. However, supplements based on carotenoid yeast have not been approved to date [94].

### 4.4. Antimycotic Properties

Feed represents one of the first and most important elements of the food chain. Preparation, composition and storage determine its quality, which has a significant impact on the health of the animals being fed. Improper preservation of raw materials for feed production or feed mixtures can result in the development of moulds (primarily storage fungi) and the consequent synthesis of mycotoxins. The most common reasons for the toxin-forming activity of fungi are feed storage temperature (above 25 °C) and improperly dried grain (moisture content > 13%) [95]. In poultry feed, fungi of the genus Penicillium [96,97,98], which produce ochratoxin, patulin and citrulline, pose a real risk to animals and are the most frequently identified [99]. Biological agents administered in a form of supplement are increasingly used to eliminate toxins from feed. These include carotenoid yeast, whose β-D-glucan which is present in the cells is an absorbent of mycotoxins. Feed supplementation with the dried carotenoid yeast *Sporidiobolus pararoseus* KM281507 contributes to a significant reduction in the mortality rate of broilers exposed to mycotoxins AFB1 (aflatoxin B1), ZEN (zearalenone), OTA (ochratoxin A) and T-2 in the diet [100]. Furthermore, the stability of the red yeast–mycotoxin complex in the gastrointestinal tract was shown to allow toxin inactivation [101].

Mycotoxins are also a general contaminant of stone fruits and their products. Fruit at the cultivation stage is infested with the genera Aspergillus, Penicillium and Alternaria contributing to economic losses and potential health risks for the consumer. Due to the thermostability of these compounds, they may be present in the end products of the fruit industry [102]. Ianiri et al. [103] investigated the ability of yeast of the genus *Sporobolomyces* to bind the patulin produced by *Penicillium expansum*. By culturing the yeast in the presence of patulin, they obtained a strain with the ability to degrade the mycotoxin. The results might suggest a potential application of yeast in the preservation of raw materials in the food industry.

### 4.5. Dietary Properties

The possibility of using yeast secondary metabolites is also indicated in the dairy industry. This sector is growing rapidly, providing consumers with new products, including functional foods. The essence of such products is scientific evidence of the health benefits of their consumption [104]. In the case of butter, researchers are investigating cholesterol lowering, the modification of fatty acid profiles or the addition of bioactive ingredients. Considering the ability of carotenoid yeast to produce lipids that include oleic acid, stearic acid and palmitic acid, the possibility of using them to create microbial butter is indicated [105].

### 4.6. Immunostimulatory Properties

Research of in vitro and in vivo models indicates that carotenoids are involved in the immune response. The effects of carotenoid fractions of microbial origin on lymphocyte proliferation or cytokine release have been investigated in the area of animal culture. For broilers, Jeong and Kim [106] reported that the addition of astaxanthin at 2.3 and 4.6 mg/kg feed had a positive effect on feed intake and weight gain during a 4-week feeding trial. Gao et al. [107] conducted an experiment investigating the effect of the addition of *P. rhodozyma* to the diet of cows during the periparturient period. Cows in the transition period experienced stress, which caused inflammation. Immune suppression can lead to teat inflammation (mastitis) or the retention of the placenta. The addition of *P. rhodozyma,* which is a source of natural astaxanthin, made it possible to reduce the levels of inflammatory factors in the cows’ serum. The authors indicated the efficacy of yeast supplementation in preventing preterm termination of pregnancy, indicating its positive role in the perinatal diet of dairy cows. However, not all study results support the efficacy of astaxanthin. Takahashi et al. [108] observed no significant differences in the feed intake and body weight gain of broilers fed diets with and without *P. rhodozyma* supplementation.

### 4.7. Supporting the Vision Process

Supporting vision in chickens is one of the bioactive properties of carotenoids [76,109]. Zhu et al. [76] analysed the effect of the addition of *Phaffia rhodozyma,* characterised by its high astaxanthin content, on production efficiency and antioxidant and immunomodulatory properties in laying hens. Somboonchai et al. [110] conducted attempts to use tofu production waste instead of soybean meal as feed for Brahman cattle. In order to take advantage of the adaptability of the yeast *Rhodotorula rubra* TISTR5134, they carried out waste fermentation with the participation of these yeasts. They achieved a low-cost feed with yeast secondary metabolites. By producing carotenoids during fermentation, *Rhodotorula* sp. provided vitamin A precursors to the diet. The application of the manufactured feed mixture influenced a higher roughage intake. However, no significant effect on meat quality or slaughter weight was shown. The authors highlighted the potential for their product to partially replace commonly used soybean meal. It would reduce farming costs while providing the necessary protein and precursors for vitamin A synthesis.

## 5. Carotenoid Bioavailability as a Key Element in the Application of Their Bioactive Properties

The characteristics indicating the bioavailability of preparations administered as nutrients or supplements should be one of the key features indicating their efficacy. In pharmacy, this concept is defined as the proportion of an ingested drug that is absorbed and available to fulfil its role in the body. Another definition refers to the fraction of an ingested nutrient that is available and used in essential physiological functions [111]. In the case of carotenoid compounds, attention is also given to the assimilation/absorption efficiency. This term refers to the proportion of carotenoids ingested and secreted into the general circulation. Consequently, a molecule present in the bloodstream can be taken up by tissues. In mammals, newly absorbed carotenoids form the so-called chylomicron carotenoids. They represent one of the measured parameters determining assimilation efficiency. The bioavailability of carotenoids is influenced by many factors. An important criterion is the form in which they occur. The selection of the appropriate steric forms of carotenoids is therefore an important part of effective supplementation, together with the observation of bioactive properties in organisms. The presence of conjugated double bonds in the carotenoid molecule influences the formation of geometric isomers. A variety of transformations influenced by physicochemical and enzymatic factors determines the formation of approximately 300 geometric isomers of β-carotene. The isomers of β-carotene which are most considered from a dietary point of view are all-trans-β-carotene and 9-cis-carotene. In plant foods or feeds, β-carotene naturally occurs in the trans form, which has a higher assimilation efficiency in mammals [112]. Experiments carried out by You et al. [112] clearly indicate a preference for the absorption of this form. This was confirmed after the administration of all-trans-β-carotene/9-cis-carotene in the form of a high-oleic safflower oil emulsion. The authors observed a higher increase in serum and in the chylomicron fraction of all-trans-β-carotene. However, a problem was the low availability of carotenoids from plant sources, which was determined to be 10–65% [113]. The reason for this is that the high fibre content and the presence of carotene–protein complexes impeded the process of carotenoid release and assimilation. An alternative could be the use of carotenoid microorganisms, which could produce the two β-carotene isomers mentioned. Moreover, their use offers the possibility of maintaining adequate concentrations of an important vitamin A precursor in the body. Increasing the production of the bioavailable and most beneficial form of carotenoid by enhancing the effect of supplementation may be optimised as early as the culture stage. Optimisation of the culture process involves the selection of physico-chemical parameters. In addition, mutagenisation of the strains should be considered, thus obtaining mutants that produce specific steric forms of the carotenoid fraction [63,114,115]. The experiment conducted by Ben-Amotz et al. [116] aimed to indicate the effect of supplementation with the algae *Dunaliella bardawil*, which is also a producer of β-carotene isomers. Prepared meals were supplemented with 40 mg of β-carotene for a period of 2 weeks. The authors indicated an increase in serum retinol levels in the study group after the supplementation period. This confirmed the bioavailability of microbial carotene. However, You et al. [112] used a higher dose of carotenoid supplementation (100 mg), observing a higher increase in β-carotene bioavailability. In the research area of red yeast utilisation studies, though, the isomeric forms of the carotenoid fractions are mostly not being analysed. The successful use of yeast carotenoids in the pharmaceutical industry will dictate their analysis. In addition to the geometric isomers mentioned, the absorption capacity of carotenoids is also influenced by the amount of fats consumed, the fibre consumed and the personal and genetic conditions of the consumer [117].

Recently, an area of intense research has been the gut microbiota and the gut–brain axis. The priority is to know and understand the interaction mechanisms taking place at the level of the microbiome and their impact on higher organism functioning. Microorganisms colonising the gut have the capacity to carry out chemical reactions. As such, they are involved in the transformation of many compounds, including polyphenols [118]. This is confirmed by studies conducted on in vitro (human cells) and in vivo (animals) models indicating their important function in carotenoid absorption. The extent of β-carotene metabolism was investigated using rats after antibiotic treatment and healthy individuals. Healthy animals accumulated less β-carotene in the liver compared to antibiotic-treated animals, indicating an effect of microflora on carotenoid metabolism. For human cells, colon faecal samples were analysed and subjected to anaerobic digestion, showing the retention of carotenoid fractions by the intestinal microflora. Only 5–50% of carotenoids are absorbed in the small intestine, with the remainder being passed into the large intestine. The efficiency of the absorption process is influenced by microbiotic diversification and the nutritional status of the gut. An experiment using an in vitro fermentation model of oral and gastrointestinal digestion versus the human intestinal microflora showed an increase in the released short-chain fatty acids (SCFAs) as fermentation time progressed. The detection of SCFAs, whose increase in concentration may have been conditioned by the presence of carotenoids, indicated the type of reactions performed by the microbiome [119]. SCFA function is particularly important in ensuring the continuity of the colonic epithelium. Disruption of this often leads to inflammatory bowel disease (IBD). Diseases with this background include Crohn’s disease [120]. Differences in the composition of gut microbiota have been detected in patients compared to healthy subjects. The differences were associated with a decrease in the relative abundance of *Firmicutes* and *Bacteroides*, and an increase in the abundance of *Proteobacteria*. An analysis of carotenoid absorption in people with Crohn’s disease showed lower fasting blood levels. Research suggests the lower assimilation of carotenoid fractions. This fact indicates the influence of quantitative and qualitative composition on the assimilation rate of carotenoids [121]. At the same time, Zhao et al. [122] found a beneficial effect of lycopene supplementation on modifying the composition of the microbiota in mice. The effect of supplementation was to promote the growth of the beneficial lactic fermentation bacteria *Bifidobacterium* and *Lactobacillus*. In contrast, the direct interaction of carotenoids with the gut microbiota is still under investigation.

## 6. Summary

Carotenoid yeasts are becoming a subject of interest for many researchers due to their ability to produce desirable secondary metabolites used in various industrial sectors. The increasing priority of consumers to consume foods based on natural ingredients is leading the food industry to develop new technologies to produce natural carotenoids.

The most advanced application research has been observed in the area of the feed industry. Many experiments confirm the effects of using whole yeast cells or pigments extracted from them. The highest commercialization of bioactive feed additives is also indicated in this area. In the food industry sector, application studies are mainly based on in vitro human cell cultures and the determination of the cytotoxicity of the extracted carotenoids. The most unexplored area is the supplementation of microbial carotenoids in humans in the pharmaceutical/medical industry. Associated with this is the need to isolate and obtain a highly purified and safe carotenoid fraction.

In this work, attention was also paid to the often-overlooked aspect of the assimilability of carotenoids produced by microorganisms. At the same time, factors influencing their assimilation have been identified. The ability of microorganisms to produce bioavailable fractions of β-carotene, which are also prevalent in plant foods, was demonstrated. As with any nutrients, carotenoids also have an impact on the gut microbiota. Furthermore, its individual composition significantly influences the bioavailability of carotenoids.

Developing an efficient and cost-effective production method is one of the first challenges in the commercialization of yeast formulations. The subsequent phases related to extraction, purification and the preservation of their bioactive properties are crucial for their application. Their application fields in various industrial sectors are still underdeveloped, and more research is required to verify yeast bioactive applications.

## Figures and Tables

**Figure 1 foods-14-01866-f001:**
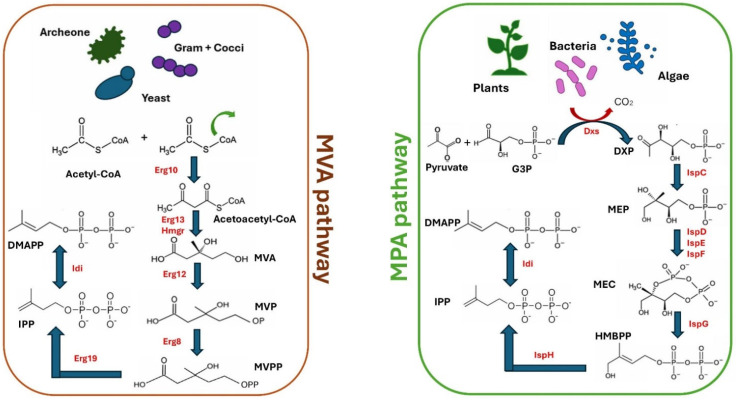
Synthesis pathways for MVA (Erg10—acetoacetyl-CoA thiolase, Erg13—3-hydroxy-3-methylglutaryl-CoA synthase, Hmgr—3-hydroxy-3-methylglutaryl-CoA reductase, MVA—mevalonate, Erg12—mevalonate kinase, MVP—mevalonate 5-phosphate, Erg8—phosphomevalonate kinase, MVPP—mevalonate-5-diphosphate, Erg19—mevalonate diphosphate decarboxylase, IPP—isopentenyl pyrophosphate, Idi—isopentenyl diphosphate isomerase, DMPPA—dimethylallyl diphosphate) and MPA (G3P—D-glyceraldhyde-3-phosphate, Dxs—1-deoxy-D-xylulose-5-phosphate synthase, DXP—1-deoxy-d-xylulose 5-phosphate, IspC—1-Deoxy-D-xylulose-5-phosphate reductoisomerase, MEP—2-C-methyl-d-erythritol 4-phosphate, IspD—4-diphosphocytidyl-2-C-methyl-D-erythritol synthase, IspE—4-diphosphocytidyl-2-C-methylerythritol kinase, IspF—2-C-methyl-D-erythritol2,4-cyclodiphosphate synthase, MEC—2-C-methyl-d-erythritol 2,4-cyclodiphosphate, IspG—4-hydroxy-3-methylbut-2-en-1-yl diphosphate synthase, HMBPP—4-hydroxy-3-methylbut-2-en-1-yl diphosphate, IspH—4-hydroxy-3-methylbut-2-enyl diphosphate reductase, IPP—isopentenyl diphosphate, Idi—isopentenyl diphosphate isomerase, DMAPP—dimethylallyl diphosphate). Based on [54,55,56,57,58].

**Figure 2 foods-14-01866-f002:**
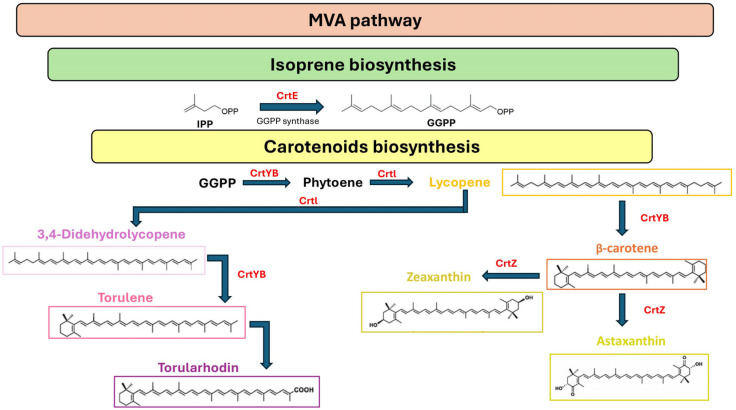
Simplified scheme of carotenogenesis in yeast (IPP—isopentenyl diphosphate, GGPP—geranylgeranyl pyrophosphate; CrtE, CrtYB, Crtl, CrtZ—genes involved in carotenogenesis). Based on: [57,58,61,62,63,64].

**Figure 3 foods-14-01866-f003:**
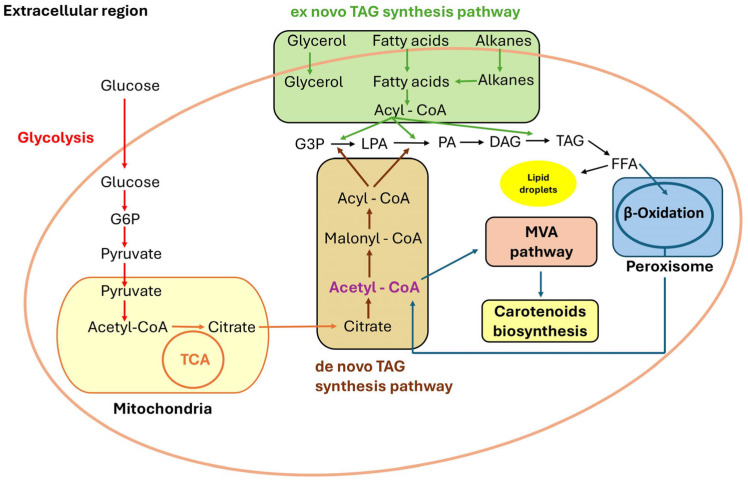
Simplified scheme of lipid synthesis correlated with carotenogenesis in oleogenic yeast (G6P—glucose-6-phosphate, TCA—tricarboxylic acid cycle, G3P—glycerol-3-phosphate, LPA—lysophosphatidic acid, PA—phosphatidic acid, DAG—diacylglycerol, TAG—triacylglyceride, FFA—free fatty acid). Based on [57,62,65,66,67].

**Table 1 foods-14-01866-t001:** The main carotenoids produced by microorganisms.

Microorganism	Type	Carotenoid Fraction Produced	Production Efficiency	References
Algae	*Haematococcus*	canthaxanthin astaxanthin	1.8 mg/g DB 22.0 mg/g *w*/*w*	[19,20]
	*Chlorella*	neoxanthin α-carotene β-carotene	0.2 μg/g 4.2 mg/g 4.3 mg/g	[21]
	*Dunaliella*	β-carotene zeaxanthin	8.3–13.5 mg 5.9 mg/g	[22,23]
Bacteria	*Micrococcus*	lycopene neoxanthin zeaxanthin β-carotene	1.5 mg/mL 2.6 mg/mL 0.9 mg/mL 0.7 g/L	[24,25,26]
	*Croceibacterium*	zeaxanthin α-carotene	412.4 μg/g 42.6 μg/g	[27]
	*Mycobacterium*	lycopene	1.4 mg/g DW 7.4 mg/g biomass	[28,29]
Yeast	*Sporobolomyces*	torulene torularodine	3.8–33.3 μg/g DW 3.1–22.9 μg/g DW	[30]
	*Sporidiobolus*	torulene torularodine β-carotene	17.0–3 1.7 mg/L 1.5–41.2 μg/g DW 10.4–18.9 mg/L	[30,31,32]
	*Phaffia*	astaxanthin β-carotene	150.0–503.6 μg/g DW 110.3–22.7 μg/g DW	[33]
	*Rhodosporidium*	β-carotene	3.3–5.1 μg/mL	[34]
	*Rhodotorula*	torulene torularodine β-carotene	20.3–95.2 μg/g DW 1.5–4.3 mg/100 g DW 38.7–77.25 μg/g DW	[35,36]
Moulds	*Blakeslea*	β-carotene	72.2–250.4 mg/L	[37]
	*Sclerotinia*	β-carotene	0.5–2.1 nmol/g DW	[38]
	*Neurospora*	β-carotene neurosporaxanthin	0.1–1.2 mg/g DB 0.2–3.7 ug/g DW	[39,40]
	*Fusarium*	neurosporaxanthin	0.02–8.3 ug/g DW	[41]
	*Mucor*	astaxanthin	9.1–11.2 g/L	[42]

Explanatory notes: DB—dry biomass; DW—dry weight.

## Data Availability

No new data were created or analyzed in this study.

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
