# Peer review of "Carotenoid Yeasts and Their Application Potential"

_foods, 2025, doi:10.3390/foods14111866_

Round 1

Reviewer 1 Report

Comments and Suggestions for Authors

General comment

The review is well-done and highlights relevant aspects of carotenoid production by yeast, as well as their areas of application. Therefore, I consider the manuscript suitable for publication in Foods journal after minor corrections.

Minor corrections:

  1. Please be consistent with the abbreviation of units throughout the manuscript. Change g/l for g/L in line 118.
  2. Please correct the phrase in lines 161-162: “In eukaryotes, archaea and Gram-positive granivores(?), biosynthesis of …”
  3. Please correct the genus scientific name Xanthophyllomuces (line 309). Must say Xanthophyllomyces.
  4. Please correct Sun e t. al. (line 312). Must say Sun et al.
  5. Is the feed supplementation of 0.5, 2.5, 12.5 % (line 313), in w/w or which units?
  6. I have a question about the blank oval in Figure 3. Is it a representation of the yeast budding process? I suggest removing it, since it doesn't specify what that component of the figure corresponds to. Or specify that it represents the budding process, if that's the intention.

With my best regards,

Author Response

Dear Reviewer 1.,

Thank you for a helpful review of our manuscript. All comments helped us to clarify our results and to improve the readability of the article. Your suggestions have been incorporated as appropriate into the revised version of the manuscript. The manuscript has been modified, reorganized, and completed. Please find below our answers. We appreciate your contribution that helps improve the manuscript.

Comments 1: Please be consistent with the abbreviation of units throughout the manuscript. Change g/l for g/L in line 118.

Response 1: Changes have been made to the line 134.

Comments 2: Please correct the phrase in lines 161-162: “In eukaryotes, archaea and Gram-positive granivores(?), biosynthesis of …”

Response 2: Corrected on cocci (line 178).

Comments 3: Please correct the genus scientific name Xanthophyllomuces (line 309). Must say Xanthophyllomyces.

Response 3: Changes have been made to the line 326.

Comments 4: Please correct Sun e t. al. (line 312). Must say Sun et al.

Response 4: Changes have been made to the line 329.

Comments 5: Is the feed supplementation of 0.5, 2.5, 12.5 % (line 313), in w/w or which units?

Response 5: Units completed- w/w  (line 330).

Comments 6: I have a question about the blank oval in Figure 3. Is it a representation of the yeast budding process? I suggest removing it, since it doesn't specify what that component of the figure corresponds to. Or specify that it represents the budding process, if that's the intention.

Response 6: Thank you for your suggestion. We agree with your opinion and have made changes to figure 3 (line 238).

Sincerely Yours,

Authors

Reviewer 2 Report

Comments and Suggestions for Authors

The manuscript on yeast carotenoids is brief, specially in the yeast part, presenting initially carotenoids in other species instead of focusing on those produced by yeasts and delving into the metabolism of yeasts as would be expected according to the title. In fact, in Figure 2 the metabolic pathways are not adequately presented, and the text that cites this figure has errors. For example, it mentions that astaxanthin is yellow but it is red, as illustrated in Figure 2 itself.  Which on the positive side is schematic in terms of the colors of carotenoids. But as already mentioned, the individual metabolic pathways in yeasts are missing.

Additionally, yeast modified to produce carotenoids such as S. cerevisiae and Y. lipolytica are omitted.

A table specifying the yield of carotenoids produced by each yeast is also required.

On the other hand, it is necessary to add information on legislative aspects, for example, there is controversy with insect carotenoids (which are currently used all over the world). But what about yeast carotenoids, what is their legislative status? And there is also a lack of information on the pathogenicity of Rhodotorulae which are opportunistic pathogens.

Author Response

Dear Reviewer 2.,

Thank you for a helpful review of our manuscript. All comments helped us to clarify our results and to improve the readability of the article. Your suggestions have been incorporated as appropriate into the revised version of the manuscript. The manuscript has been modified, reorganized, and completed. Please find below our answers. We appreciate your contribution that helps improve the manuscript.

Comments 1: The manuscript on yeast carotenoids is brief, specially in the yeast part, presenting initially carotenoids in other species instead of focusing on those produced by yeasts and delving into the metabolism of yeasts as would be expected according to the title.

Response 1: The premise of the study was to analyse the application potential of carotenoid yeasts as the title indicates. The first chapters were intended to introduce the reader to the topic of carotenoids, allowing for an easier understanding of the subsequent content related to their application in different industrial sectors.

Comments 2: In fact, in Figure 2 the metabolic pathways are not adequately presented, and the text that cites this figure has errors. For example, it mentions that astaxanthin is yellow but it is red, as illustrated in Figure 2 itself.  Which on the positive side is schematic in terms of the colors of carotenoids. But as already mentioned, the individual metabolic pathways in yeasts are missing.

Response 2: Thank you for drawing attention to this issue. We have been changed colour of astaxanthin on the figure 2 line 214.

The premise of this diagram was to show a simplified pathway for carotenoid synthesis (which is included in the name of figure 2). Carotenogenesis is a complex process involving many chemical transformations. However, this review focuses on the application nature of carotenoids hence a simplified scheme is presented to enable any reader to understand the process.

Comments 3: Additionally, yeast modified to produce carotenoids such as S. cerevisiae and Y. lipolytica are omitted.

Response 3: S. cerevisiae and Y. lipolytica do not belong to this group of carotenoid yeasts. The synthesis of carotenoids by the yeast types is only possible through genetic engineering techniques - the authors mentioned this aspect in the line 60-61.

Comments 4: A table specifying the yield of carotenoids produced by each yeast is also required.

Response 4: The yield of carotenoids produced by representatives of carotenoid yeast, but also by other microorganisms, has been added as column 3 in Table 1 (line 121). Production efficiency is highly variable depending on the genus, species, strain, culture conditions and culture medium. Therefore, yields were assigned to the specific yeasts analysed by the researchers.

Comments 5: On the other hand, it is necessary to add information on legislative aspects, for example, there is controversy with insect carotenoids (which are currently used all over the world). But what about yeast carotenoids, what is their legislative status? And there is also a lack of information on the pathogenicity of Rhodotorulae which are opportunistic pathogens.

Response 5: The topic on legal aspects, which was already mentioned in the introduction (line 41-53), was developed. The possibility of opportunistic pathogens in the genus Rhodotorula was also mentioned (line 61-62).

Thank you for highlighting a topic related to insects. We agree that this topic is controversial when they are used as food additives. We mentioned this aspect in the line 47-50.

Sincerely Yours,

Authors